# ECG Classification Based on Wasserstein Scalar Curvature

**DOI:** 10.3390/e24101450

**Published:** 2022-10-11

**Authors:** Fupeng Sun, Yin Ni, Yihao Luo, Huafei Sun

**Affiliations:** School of Mathematics and Statistics, Beijing Institute of Technology, Beijing 100081, China

**Keywords:** ECG classification, positive definite symmetric matrix manifold, local statistics, Wasserstein metric, curvature

## Abstract

Electrocardiograms (ECG) analysis is one of the most important ways to diagnose heart disease. This paper proposes an efficient ECG classification method based on Wasserstein scalar curvature to comprehend the connection between heart disease and the mathematical characteristics of ECG. The newly proposed method converts an ECG into a point cloud on the family of Gaussian distribution, where the pathological characteristics of ECG will be extracted by the Wasserstein geometric structure of the statistical manifold. Technically, this paper defines the histogram dispersion of Wasserstein scalar curvature, which can accurately describe the divergence between different heart diseases. By combining medical experience with mathematical ideas from geometry and data science, this paper provides a feasible algorithm for the new method, and the theoretical analysis of the algorithm is carried out. Digital experiments on the classical database with large samples show the new algorithm’s accuracy and efficiency when dealing with the classification of heart disease.

## 1. Introduction

As one of the most common and deadly diseases in the world, heart disease poses a significant threat to people’s happiness and healthy life. With the increase of social pressure, the incidence of heart disease is proliferating in recent years [1]. Therefore, it is of great indispensability to realize efficient diagnosis, real-time monitoring and prediction for heart disease. At present, the diagnosis is mainly made through doctors’ manual analysis of ECG (electrical activity records of patient’s heart contraction) [2]. With no exception, ECG analysis requires doctors to command professional and detailed medical knowledge. However, as a kind of valuable medical resource, the distribution of experienced doctors is not well balanced in different regions of the world. Thus, the life and health of patients can not be fully protected in a situation with poor medical conditions and resources.

With the rapid development of computer technology, using computer-aided diagnosis technology to screen heart diseases has become a new technical method to alleviate the imbalance of artificial medical resources. The normal ECG, that is healthy ECG, shows regular changes in the PQRST complex. PQRST complex  [3] is the pattern of the electrical activity of the heart during one cardiac cycle as recorded by ECG, including P–R interval, QRS complex, Q–T interval and S–T segment. Furthermore, different pathological features will cause symmetry to vanish. The computer-aided diagnosis techniques aim to extract different features of ECG and detect fluctuation of the PQRST complex in view of these features to achieve accurate classification.

Commonly used computer-aided diagnosis techniques mainly extract features from the viewpoints of signal analysis, dynamic system modeling (DSA) and topological data analysis (TDA), which are also combined with classic statistical analysis [4] and machine learning [5,6,7,8,9,10]. Signal analysis can directly extract morphological features such as amplitude [11,12] or use the wavelet transform to acquire frequency domain features [13,14]. Other emerging signal analysis methods [15] also have the potential to analyze ECG. Dynamical system analysis is used to model cardiac dynamical system through phase space reconstruction [16,17,18,19,20]. Topological data analysis transforms ECG signals into point clouds by time-delay embedding or Fourier transform and then extracts topological features from point clouds [21,22,23,24,25,26]. Above mentioned methods are adaptive with different applied conditions. Despite various advantages of conventional methods, most of them are limited by small sample size or poor interpretation. Thus, a new way without such limitations needs to be proposed.

Inspired by persistent homology (PH) [27] in TDA, a geometric data analysis method is proposed and applied to ECG classification. PH transforms the original ECG signals into point clouds in Euclidean space by embedding them in a suitable way and extracting topological features from point clouds. Topological features can reflect different pathological characteristics of the original ECG, so as to realize the classification. However, the topological characteristics cannot focus on fine local information of point clouds, also the category number and sample size are limited. Thus, it is necessary to describe the local differences between point clouds in a more accurate method.

Each point of the point cloud in *n* dimensional Euclidean obtained by short-time Fourier Fast transform (STFFT) [26] is mapped to a *n* dimensional Gaussian distribution by local statistics, that is, the point cloud in Euclidean space is transformed into the point cloud on the positive definite symmetric matrix manifold SPD(n). SPD(n) becomes a Riemannian manifold after being endowed with the natural Riemannian metric induced by Wasserstein distance [28]. We reveal the local differences of point clouds on SPD(n) [29] by Wasserstein scalar curvature (WSC). By analyzing the Wasserstein scalar curvature histogram (WSCH) of the converted point clouds, we originally defined a new data characteristic, called Wasserstein scalar curvature dispersion (WSCD) which can express the pathological changes of heart. Finally, the ECG classification algorithm based on Wasserstein scalar curvature (WSCEC) is designed to classify ECG signals.

This paper is organized as follows. Section 2 gives the preliminaries. Section 3 presents WSCEC algorithm. In Section 4, we analyze the numerical results. Finally, we discuss the main conclusions and prospects for a future research in Section 5.

## 2. Preliminaries

In this section, we will introduce some basic preliminaries such as STFFT, local statistics and Wasserstein geometric structure on SPD(n).

### 2.1. Short-Time Fourier Fast Transform

Fourier transform decomposes signals into waves with different frequencies and reveals certain features hidden in the time domain. For discrete inputs, fast Fourier transform (FFT) is a widely used tool. Given a signal $T={\left\{t_{i}\right\}}_{i=1}^{n}$ with even *n*, $t_{i}$ can be represented as:(1)$$t_{i}=\frac{1}{n}\sum_{k=0}^{n-1}C_{k}e^{j\frac{2\pi }{n}ki},$$
where $C_{k}=\sum_{i=0}^{n-1}t_{i}e^{-j\frac{2\pi }{n}ki}$.

Let $C_{k}=a_{k}+b_{k}j$, then we have $C_{n-k}=a_{k}-b_{k}j$ and
(2)$$C_{k}e^{j\frac{2\pi }{n}ki}+C_{n-k}e^{j\frac{2\pi }{n}\left(n-k\right)i}=2a_{k}cos\left(\frac{2\pi ki}{n}\right)-2b_{k}sin\left(\frac{2\pi ki}{n}\right)=A_{k}\cos\left(\frac{2\pi ki}{n}+\phi_{k}\right),$$
where $A_{k}=2\sqrt{a_{k}^{2}+b_{k}^{2}}$, $\tan\left(\phi_{k}\right)=\frac{b_{k}}{a_{k}}$.

If the sample frequency is $f_{s}$, by combining Equations (Equation 1) and (Equation 2), we have
(3)$$t\left(s\right)=\frac{1}{n}\sum_{k=0}^{\frac{n}{2}-1}A_{k}\cos\left(f_{s}\frac{2\pi ks}{n}+\phi_{k}\right),$$
where $t_{i}=t\left(\frac{i}{f_{s}}\right)$.

Now we use FFT and sliding windows (short-time Fourier transform) to convert signal $T={\left\{t_{i}\right\}}_{i=1}^{n}$ to a point cloud in $R^{d}$ where *d* is odd. Let *l* be window length and τ be sliding speed. Firstly, we transform *T* into a signal set $P_{l,\tau }\left(T\right)={\left\{p_{k}\right\}}_{k=1}^{\hat{n}-1}$ with $\hat{n}=\left\lfloor \frac{n-l}{\tau }\right\rfloor$ and $p_{k}=\left(t_{k\tau },t_{k\tau +1},\cdots ,t_{k\tau +(l-1)}\right)$, where ⌊·⌋ denotes floor operator. Secondly, we transform each $p_{k}$ into a point in $R^{d}$ by choosing the first *d* bases from FFT and obtain the following point cloud:(4)$$T\to P_{l,\tau }\left(T\right)={\left\{p_{k}\right\}}_{k=1}^{\hat{n}-1}\to S_{F}\left(T\right)=S_{l,\tau ,d}\left(T\right),$$
where $S_{l,\tau ,d}\left(T\right)={\left\{\frac{2}{l}\left(a_{0}^{k},a_{1}^{k},b_{1}^{k},\cdots ,a_{\frac{d-1}{2}}^{k},b_{\frac{d-1}{2}}^{k}\right)\right\}}_{k=1}^{\hat{n}-1}$ and d≤l.

### 2.2. Local Statistic

Objective phenomena in nature are often disturbed by many small random variables, which assign a random distribution with the local neighborhood of any point in the point cloud. If the factors which affect the local distribution of a point cloud are considered small and complex enough, then according to the law of large numbers and the central limit theorem, we can assume that the local statistics come from a high-dimensional Gaussian distribution whose parameters are neighborhood mean and neighborhood covariance matrix.

In data science, *k*-nearest neighbors (kNN) algorithm algorithm provides a natural neighborhood selection method. The idea is to search for the nearest *k* points as the neighborhood samples of any fixed point in the point cloud. To acquire local statistics, for every point $P_{i}\in S_{F}\left(T\right)$, we can search kNN to obtain neighborhood $N_{i}=\left\{N_{ij}|j=1,\cdots ,k\right\}$, and calculate local mean $\mu_{i}$ and covariance matrix $\Sigma_{i}$:(5)$$\mu_{i}=\frac{1}{k}\sum_{j=1}^{k}N_{ij},\;\Sigma_{i}=\sum_{j=1}^{k}{\left(N_{ij}-\mu_{i}\right)}^{T}\left(N_{ij}-\mu_{i}\right).$$

Consequently, $S_{F}\left(T\right)$ is converted to a point cloud in SPD(d):(6)$$S_{F}\left(T\right)\to S_{P}\left(T\right)=S_{SPD}\left(T,k\right)=\left\{\Sigma_{i}|P_{i}\in S_{F}\left(T\right)\right\}.$$

### 2.3. Wasserstein Geometric Structure on SPD(n)

Wasserstein distance describes the minimal energy used to transport one distribution to another. It can be used to measure the difference between two distributions and is vividly called earth-moving distance [28,30,31,32,33].

Let $F_{1},F_{2}$ be two distributions. Then the Wasserstein distance between $F_{1}$ and $F_{2}$ based on the $L^{p}$ norm is defined as the infimum of geodesic distance integral needed for transporting probability measure element [34]:(7)$$W_{p}\left(F_{1},F_{2}\right)=\underset{p∼\Pi \left(F_{1},F_{2}\right)}{\inf}{\left(E_{\left(x,y\right)∼p}{[∥x-y∥}^{p}]\right)}^{\frac{1}{p}},$$
where $\Pi \left(F_{1},F_{2}\right)$ is the set of joint distributions taking $F_{1},F_{2}$ as marginal distributions and *E* is the expectation.

Although the definition is abstract and there is no explicit expression for the general Wasserstein distance, the Wasserstein distance based on the $L^{2}$ norm between any two Gaussian distributions in $R^{n}$ has the following explicit expression [35]:(8)$$W\left(N_{1},N_{2}\right)=∥\mu_{1}-\mu_{2}∥+{\left(\mathrm{tr}\left(\sum_{1}+\sum_{2}-2{\left(\sum_{1}\sum_{2}\right)}^{\frac{1}{2}}\right)\right)}^{\frac{1}{2}},$$
where $\mu_{i}$ and $\sum_{i}$ are the mean and covariance matrix of Gaussian distribution $N_{i}$, i=1,2.

Wasserstein distance can be induced by a Riemannian metric on SPD(n) defined as:(9)$$g_{W}{|}_{S}\left(X,Y\right)=\frac{1}{2}\mathrm{tr}\left(\Gamma_{S}\left[Y\right]X\right),$$
where $S\in SPD\left(n\right)$, $X,Y\in T_{S}SPD\left(n\right)$ are tangent vectors and $\Gamma_{S}\left[Y\right]$ is the solution of Sylvester equation $S\Gamma_{S}\left[Y\right]+\Gamma_{S}\left[Y\right]S=Y$ [36].

Because the geodesic distance induced by Equation (Equation 9) is consistent with the original definition of Wasserstein distance (Equation 8), we call $g_{W}$ Wasserstein metric. In addition, we write the Riemannian manifold SPD(n) endowed with Wasserstein metric as $\left(SPD\left(n\right),g_{W}\right)$.

For any $S\in (SPD\left(n\right),g_{W})$, let X,Y be the smooth vector field of $SPD\left(n\right)$. P.A. Absil et al. provide the explicit expression of Riemannian curvature tensor $\langle R_{XY}X,Y\rangle$ at *S*  [29,37]:(10)$$\begin{array}{cc} R\left(X,Y,X,Y\right)=3\mathrm{tr} & \left(\Gamma_{S}\left[X\right]S\Gamma_{S}\left(\Gamma_{S}\left[X\right]\Gamma_{S}\left[Y\right]-\Gamma_{S}\left[Y\right]\Gamma_{S}\left[X\right]\right)S\Gamma_{S}\left[Y\right]\right). \end{array}$$

Furthermore, the scalar curvature at *S* satisfies
(11)$$\begin{array}{cc} \rho \left(S\right) & =\sum_{i=1}^{n}\sum_{j=1}^{n}R\left(e_{i},e_{j},e_{i},e_{j}\right)\\ \; & =3\mathrm{tr}\left(U\Lambda \left(U+U^{T}\right)+\left(U+U^{T}\right)\Lambda U+\left(U+U^{T}\right)\Lambda U\Lambda \left(U+U^{T}\right)\right), \end{array}$$
where $\left\{e_{i}\right\}$ is any standard orthonormal basis of $T_{S}SPD(n)$, $\Lambda =\mathrm{diag}\left(\lambda_{1},\cdot \cdot \cdot ,\lambda_{n}\right)$ is orthogonally similar to *S*, $\lambda_{i}$ is the *i*th eigenvalue of *S*, and $U={\left[\frac{1}{\lambda_{i}+\lambda_{j}}\right]}_{i<j}$ is an upper triangular matrix.

Note that the scalar curvature of *S* can be controlled by the second small eigenvalue of *S*. In fact, there exists a standard orthonormal basis $\left\{e_{k}\right\}$ of $T_{S}SPD\left(n\right)$, such that ∀$e_{k_{1}},e_{k_{2}}\in \left\{e_{k}\right\}$,
(12)$$0<K_{S}\left(e_{k_{1}},e_{k_{2}}\right)=\sum_{j=1}^{n}R\left(e_{k_{1}},e_{k_{j}},e_{k_{2}},e_{k_{j}}\right)<\frac{3}{\lambda_{min2}\left(S\right)},$$
where $\lambda_{min2}$ is the second small eigenvalue of *S*.

Furthermore, by Equation (Equation 11), we have
(13)$$0<\rho \left(A\right)<\frac{3n(n-1)}{\lambda_{min2}\left(S\right)}.$$

The boundedness of WSC indicates that the curvature of the local covariance matrix is controllable unless the local covariance degenerates in two dimensions or more. Consequently, for most neighborhoods, as long as using appropriate embedding methods to make sure there is no degeneration beyond two dimensions, the WSC of point cloud on SPD(n) will be in a controllable range, which provides a theoretical criterion for the robustness of our algorithm.

## 3. ECG Classification Algorithm Based on Wasserstein Scalar Curvature

In this section, we will introduce an ECG classification algorithm based on Wasserstein scalar curvature (WSCEC) which can detect heart disease. Wasserstein scalar curvature dispersion is extracted to reveal the change of regularity of the PQRST complex. The framework of the WSCEC algorithm is as follows.

Continuous ECG signals are segmented and denoised by interpolation and filter to obtain multiple single heartbeats.Every single ECG is transformed into a point cloud in Euclidean space by STFFT. Through local statistics, the point cloud in Euclidean space is converted to the point cloud on SPD(n).Calculate the WSC of each point to obtain WSCH and extract WSCD as the feature.Do auxiliary diagnosis according to clustering results.

The intuitive algorithm pipeline is shown in Figure 1.

### 3.1. Preprocessing of ECG Signal

We adopt the idea of Butterworth filter algorithm [38] to cut off noisy portions with spectral power over 50 Hz. By developing a local search algorithm to find the periodic R-peak in the PQRST complex, we transform continuous ECG signals into multiple single heartbeats with a length of 300.

Noting that the length of the sharp part of the QRS complex in normal ECG is around 10, we set window length l=10 and sliding speed τ=1 to emphasize the change of the QRS complex when sliding the window. Then by using Equation (Equation 4) and taking d=3, we convert various single heartbeats to point clouds in $R^{3}$. Let $T_{s}$ be a standard normal ECG signal and denote $S_{F}\left(T_{s}\right)$ as the Euclidean point cloud of $T_{s}$ after STFFT. Figure 2 shows $S_{F}\left(T_{s}\right)$ intuitively.

In an attempt to describe the local differences of Euclidean point clouds more accurately, we obtain neighborhood properties by local statistics. We combine kNN algorithm with Equation (Equation 5) to obtain $S_{P}\left(T_{s}\right)$, and the parameter *k* of kNN algorithm is 20. With an attempt to give readers an intuitive understanding of the structure of point clouds on SPD(3), we acquire Wasserstein distance matrices by Equation (Equation 8) and present the grayscale images of Wasserstein distance matrice for $T_{s}$, see Figure 3.

### 3.2. Feature Extraction

The pathological differences of original ECG signals are completely reflected by the local information of point clouds, and the differences of local information are further reflected in the neighborhood mean and covariance matrix, that is, reflected in the point-by-point difference of point clouds on SPD(n). Since Wasserstein scalar curvature reflects the structural relationship between a point and its adjacent points, we can use Wasserstein scalar curvature (WSC) to describe different pathological features of ECG signals.

Firstly, we calculate WSC of each point in the distribution point cloud, then the corresponding WSC sequence $W={\left\{w_{i}\right\}}_{i=1}^{\hat{n}}$ can be obtained. Furthermore, given *m* and *b*, we can acquire Wasserstein scalar curvature histogram (WSCH):(14)$$H\left(m,b\right)=\left\{\left(\left[mj,m(j+1)\right),y_{j}\right)\;|\;0\leq j\leq \left\lfloor \frac{b}{m}\right\rfloor \right\},$$
where
(15)$$y_{j}=|\left\{w_{k}\in W|mk\leq w_{k}<m\left(k+1\right)\right\}|,$$
⌊·⌋ denotes the integer operator and |·| denotes the cardinality or size of a finite set.

Figure 4 performs the WSCH of the point cloud $S_{P}\left(T_{s}\right)$ on SPD(3) where m=10, b=200. Now we give the definition of Wasserstein scalar curvature dispersion (WSCD) to describe the differences of the WSC sequence.

**Definition** **1.**
*For the given WSC sequence W, histogram H(m,b) and $0\leq s\leq \left\lfloor \frac{b}{m}\right\rfloor \;\left(s\in N_{+}\right)$, we define:*

(16)
$$\begin{array}{cc} cur_{1}\left(m,b,s\right) & =median\;of\;U_{1},\\ cur_{2}\left(m,b,s\right) & =\frac{1}{|U_{2}|-s-1}\sum_{j\geq s+1}{\left(y_{j}-\frac{\sum_{j\in U_{2}}y_{j}}{\left\lfloor \frac{b}{m}\right\rfloor }\right)}^{2},\\ cur(m,b,s) & =(cur_{1}\left(m,b,s\right),cur_{2}\left(m,b,s\right)), \end{array}$$

*where*

(17)
$$\begin{array}{cc} U_{1} & =\left\{w_{k}\in W|ms\leq w_{k}\leq b\right\},\\ U_{2} & =\left\{k\geq s+1|y_{k}\neq 0\right\}. \end{array}$$

*We call $cur_{1}\left(m,b,s\right)$ Wasserstein scalar curvature transverse dispersion of W, $cur_{2}\left(m,b,s\right)$ Wasserstein scalar curvature longitudinal dispersion of W and cur(m,b,s) Wasserstein scalar curvature dispersion of W, respectively.*


In histogram H(m,b), transverse dispersion $cur_{1}\left(m,b,s\right)$ is the median of the intersection between W and [ms,b]. $cur_{1}\left(m,b,s\right)$ describes the homogeneity of W ranging in [m(s+1),b] horizontally.

The longitudinal dispersion $cur_{2}\left(m,b,s\right)$ represents the fluctuation of the column, which can be regarded as the correction of the standard deviation. If the column heights of some curvature intervals in H(m,b) are 0, then this fluctuation is further amplified. $cur_{2}\left(m,b,s\right)$ describes the uniformity of W ranging in [m(s+1),b] vertically. In particular, $cur_{2}\left(m,b,0\right)$ is the standard deviation of the curvature histogram H(m,b) if the columns are all nonzero.

Therefore, cur(m,b,s) describes the homogeneity of W overall, and cur(m,b,s) is closer to $\left(\frac{1}{2}\left(b+ms\right),0\right)$ if the columns are more evenly distributed. Since the ECG of healthy heartbeats has strong regularity, their WSCD is more even, as shown in Figure 5. We consider WSCD cur(m,b,s) as the feature of our final classification.

### 3.3. Case Analysis

Figure 6 shows seven types of single heartbeats. Compared with normal ECG signals, the other six diseases have different effects on the PQRST complex. The QRS complex of left bundle branch block heartbeats (L.B.B.B.) or right bundle branch block heartbeats (R.B.B.B.) is obviously broadened, and generally, there are two R peaks. The P wave of atrial premature heartbeats (A.P.) occurs earlier and is significantly different from that of the sinus. P.V.C. has the larger QRS complex amplitude, which always companies with more significant range differences in the waves. The QRS complex in the fusion of ventricular and normal heartbeats (F.V.N.) is the fusion of normal heartbeat and ventricular flutter heartbeats (V.F.), and its deterioration will change into V.F. whose waveform is similar to a sine wave, in which case cardiopulmonary resuscitation is needed for treatment.

Figure 7 shows the point clouds in $R^{3}$ for ECG signals with six pathological features. Combining Figure 2 with Figure 7, the differences of waves are reflected in the differences of local information of point clouds in $R^{3}$. Notice that the local information of the point clouds of ECG signals with diseases is significantly different from those of normal ECG signals except A.P., this may be due to the fact that A.P. is generated by atrial abnormal excitation foci in advance, and sometimes there are only *P* wave differences with normal ECG signals. Therefore, their local structures of Euclidean point clouds are similar.

There are also similarities among the point cloud structures of P.V.C., F.V.N., and V.F., which may be due to the fact that these three types of diseases are also generated from ventricular ectopic excitation foci, their pathogenesis and trend also have a certain progressive relationship. The similarities of local structures between different ECG signal point clouds also reflect the necessity of introducing WSC to describe the differences of such fine structures more accurately.

By local statistics, we change the point clouds of ECG signals with diseases into the point clouds on SPD(3). The grayscale images of Wasserstein distance matrices describe the dispersion of distribution point clouds on SPD(3), where black represents the zero distance and white represents the maximum distance. Local structure differences of distribution point clouds can be visually presented in Figure 3 and Figure 8.

We calculate the WSC of each point to precisely characterize the differences in neighborhood information of different point clouds on SPD(3). WSCHs are also formed, as shown in Figure 9.

WSC sequences corresponding to seven kinds of ECG signals are almost located at [0, 200], as shown in Figure 4 and Figure 9. By Equation (Equation 13), it can be inferred that the neighborhood information of most points in the Euclidean point clouds does not have more than two-dimensional degradation, which shows the effectiveness of the STFFT method and the selected parameters.

In the histogram of normal ECG signal and A.P., their columns are evenly distributed horizontally and the columns of A.P. fluctuate more violently. The histograms of P.V.C., F.V.N. and V.F. are less evenly distributed horizontally and their fluctuation of columns is similar. The columns of L.B.B.B. and R.B.B.B. are concentrated in the smaller part of the WSC values. To identify these seven ECG signals more accurately, we calculate their WSCD.

We denote the Wasserstein scalar curvature dispersion plane as:$$D=\left\{\left(cur_{1}\left(m,b,s\right),cur_{2}\left(m,b,s\right)\right)|cur_{i}\left(m,b,s\right)\in R,i=1,2\right\}.$$

Figure 10 shows the calculation results of WSCD for seven kinds of ECG signals and every type is sampled 50 signals. Transverse curvature dispersion and longitudinal curvature dispersion reveal the length and the fluctuation of the QRS complex, respectively. The transverse curvature dispersion gets larger with an increasing width of the QRS complex and the longitudinal curvature dispersion turns bigger with more violent fluctuation of the QRS complex.

QRS complex in normal ECG signal and A.P. are most regular and their transverse curvature dispersion is both largest. Notice that the lesion in the atria causes some changes in the QRS complex, hence the longitudinal curvature dispersion of A.P. is higher than normal heartbeats. Let $D_{0}=\left(25,200\right]\times \left[0,25\right]$ denote normal status of heart and $D_{1}=\left(25,90\right]\times \left(25,+\infty \right)$ denote atrial abnormal.QRS complexes in V.F., F.V.N. and P.V.C. are not so wide and their fluctuation gradually expands, hence the transverse curvature dispersion of these three kinds of heartbeats is moderate and their longitudinal curvature dispersion becomes bigger and bigger. Since these three kinds of heartbeats are caused by ventricular lesions, their longitudinal dispersion has some similarities. Let $D_{21}=\left(10,25\right]\times \left[0,50\right]$ denote V.F., $D_{22}=\left(10,25\right]\times \left[40,70\right]$ denote V.F. N. and $D_{23}=\left(10,25\right]\times \left[60,+\infty \right)$ denote P.V.C. Furthermore, let $D_{2}=\left(10,25\right]\times \left[0,+\infty \right)$ denote ventricular abnormal.QRS complex in L.B.B.B. and R.B.B.B. are usually widest, hence transverse curvature dispersion is smallest. In addition, the existence of two R peaks in R.B.B.B. is more obvious than that in L.B.B.B., which results in more fluctuation in QRS and a larger longitudinal curvature dispersion. Let $D_{31}=\left[0,10\right)\times \left[0,140\right]$ denote L.B.B.B. and $D_{32}=\left[0,10\right)\times \left[100,\infty \right)$ denote R.B.B.B.. In addition, let $D_{3}=\left[0,10\right)\times \left[0,+\infty \right)$ denotes bundle branch block area.

For those heartbeats that land in $D_{4}=D-\cup_{j=0}^{3}D_{j}$, we consider these heartbeats are from other abnormal areas. Thus, we derive a symptom description domain partition $D=\cup_{j=0}^{4}D_{j}$. Given an ECG signal $T_{i}$, the auxiliary diagnostic analysis of heart disease is as follows:If $cur_{T_{i}}\left(m,b,s\right)\in D_{0}$, we deem $T_{i}$ is normal.If $cur_{T_{i}}\left(m,b,s\right)\in D_{1}$, we consider $T_{i}$ is A.P..If $cur_{T_{i}}\left(m,b,s\right)\in D_{2}\cap D_{21}-D_{22}\cup D_{23}$, we consider $T_{i}$ is V.F.. If $cur_{T_{i}}\left(m,b,s\right)\in D_{2}\cap D_{22}-D_{21}\cup D_{23}$, we think $T_{i}$ is V.F. N.. If $cur_{T_{i}}\left(m,b,s\right)\in D_{2}\cap D_{23}-D_{21}\cup D_{22}$, we consider $T_{i}$ is P.V.C.. If $cur_{T_{i}}\left(m,b,s\right)\in D_{2j}\cap D_{2k},\;1\leq j,k\leq 3$, we think $T_{i}$ has the pathological features of both $D_{2j}$ and $D_{2k}$. In this case, we cannot classify $T_{i}$, but we can label it ventricular abnormal.If $cur_{T_{i}}\left(m,b,s\right)\in D_{3}\cap D_{31}-D_{32}$, we consider $T_{i}$ is L.B.B.B.. If $cur_{T_{i}}\left(m,b,s\right)\in D_{3}\cap D_{32}-D_{31}$, we think $T_{i}$ is R.B.B.B.. If $cur_{T_{i}}\left(m,b,s\right)\in D_{31}\cap D_{32}$, we cannot classify $T_{i}$, but we can label it bundle branch block.If $cur_{T_{i}}\left(m,b,s\right)\in D_{4}$, $T_{i}$ cannot be classified.

Now we give our algorithm to show how to classify the ECG signals and carry out the auxiliary diagnosis.

### 3.4. WSCEC Algorithm

Let $T={\left\{T_{i}\right\}}_{i=1}^{N}=\cup_{j=0}^{r}T_{j}=\cup_{j=0}^{4}Q_{j}$ be the set of the given ECG signal where $T_{0}$ is the set of normal ECG in T, $T_{j}$ (1≤j≤r) is the set of ECG with *j*th pathological feature in T, and $Q_{j}$ is the original heartbeat set which is corresponding to symptom description domain $D_{j}$. Then WSCEC algorithm is shown in Algorithm 1.
**Algorithm 1** WSCEC algorithm**Input:** ECG set T; parameter k,m,s,ϵ**Output:** Classification result $T=\cup_{j=0}^{r+1}{\tilde{T}}_{j}\cup {\tilde{Q}}_{4}=\cup_{j=0}^{4}{\tilde{Q}}_{j}$1:Choose standard normal ECG signal $T_{s}$2:For every ECG signal $T_{i}$ in T, acquire point cloud $S_{F}\left(T_{i}\right)$ after STFFT by Equation (Equation 4), where l=10, τ=1 and d=33:Acquire point cloud $S_{SPD}\left(T_{i},k\right)$ of $T_{i}$ and $S_{SPD}\left(T_{s},k\right)$ of $T_{s}$ by kNN algorithm and Equation (Equation 5)4:Calculate scalar curvature at each point in $S_{SPD}\left(T_{s},k\right)$ by Equation (Equation 11), take *b* as the minimum of the max scalar curvature and $\frac{3d(d-1)}{\epsilon }$5:Calculate scalar curvature at each point in $S_{SPD}\left(T_{i},k\right)$ by Equation (Equation 11) and obtain curvature histogram H(m,b) by Equation (14)6:Calculate Wasserstein scalar curvature dispersion $cur_{T_{i}}\left(m,b,s\right)$ of $T_{i}$ by Equation (Equation 16)7:Give the classification result $T=\cup_{j=0}^{r+1}{\tilde{T}}_{j}\cup {\tilde{Q}}_{4}=\cup_{j=0}^{4}{\tilde{Q}}_{j}$ by symptom description domain partition $D=\cup_{j=0}^{4}D_{j}$8:**Output:**$T=\cup_{j=0}^{r+1}{\tilde{T}}_{j}\cup {\tilde{Q}}_{4}=\cup_{j=0}^{4}{\tilde{Q}}_{j}$

For the output of Algorithm 1, ${\tilde{T}}_{0}$ represents the healthy heartbeat set, ${\tilde{T}}_{j}$ (1≤j≤r) represents the heartbeat set with *j*th pathological feature, and ${\tilde{T}}_{r+1}$ represents the unclassified heartbeat set with label of lesion area. ${\tilde{Q}}_{j}$ denotes the classified heartbeat set which is corresponding to the symptom description domain $D_{j}$. Thus, for an unclassified heartbeat $T_{i}\in {\tilde{T}}_{r+1}$, although we cannot know the exact type of $T_{i}$, we can also know which part of the heart is abnormal.

## 4. Digital Experiment

In this section, the main numerical results are introduced. The sampled data comes from MIT-BIH Arrhythmia Database [39]. We sample 5000 heartbeats with distinguishing features, including 2500 normal heartbeats (N), 1000 premature ventricular contraction heartbeats (P.V.C.), 450 left bundle branch block heartbeats (L.B.B.B.), 450 right bundle branch block heartbeats (R.B.B.B.), 200 atrial premature heartbeats (A.P.), 200 fusion of ventricular and normal heartbeats (F.V.N.) and 200 ventricular flutter heartbeats (V.F.). (The results of the WSCEC algorithm are presented and the comparison with Convolution Neural Network (CNN) method and persistent homology (PH) method are discussed (see https://github.com/bitni2609/bit_curvature_ECG (accessed on 14 September 2022) for the sampled data and algorithms).

### 4.1. Results of WSCEC Algorithm

Figure 11 shows the calculation results of WSCD for the 5000 segment ECG signals sampled, the parameters are $\epsilon =0.09,\;b=\frac{3d(d-1)}{\epsilon }=200,\;m=1$, and s=0. Note that there are overlaps between bundle branch block heartbeats and abnormal ventricular heartbeats such as F.V.N. and P.V.C., this may be because all these heartbeats have widened QRS complex and some of them are indeed pretty similar. We introduce sensitivity or true positive rate (TPR), noise removal rate (NRR), precision or positive predictive value (PPV) and $F_{1}$ score to show the efficiency of our symptom description domain partition.

Let $T=\cup_{j=0}^{4}Q_{j}$ be the original ECG set and $T=\cup_{j=0}^{4}{\tilde{Q}}_{j}$ be the set after classification. Then for all 0≤j≤4,
(18)$$\begin{array}{cc} {\mathrm{TPR}}_{j} & =\frac{|{\tilde{Q}}_{j}\cap Q_{j}|}{|Q_{j}|},\;{\mathrm{NRR}}_{j}=1-\frac{|{\tilde{Q}}_{j}\cap \cup_{k\neq j}Q_{k}|}{|\cup_{k\neq j}Q_{k}|},\\ {\mathrm{PPV}}_{j} & =\frac{|{\tilde{Q}}_{j}\cap Q_{j}|}{|{\tilde{Q}}_{j}|},\;F_{1}\;score=2\frac{|{\tilde{Q}}_{j}\cap Q_{j}|}{|Q_{j}\left|+\right|{\tilde{Q}}_{j}|}=2\frac{{\mathrm{PPV}}_{j}\times {\mathrm{TPR}}_{j}}{{\mathrm{PPV}}_{j}+{\mathrm{TPR}}_{j}}, \end{array}$$
where |·| denotes the cardinality or size of a finite set.

${\mathrm{TPR}}_{j}$ describes the accuracy of original ECG signals with lesion area *j* preserved by the new classification. ${\mathrm{NRR}}_{j}$ represents the success rate of removing ECG signals except *j*. ${\mathrm{PPV}}_{j}$ describes the proportions of a positive result and $F_{1}$ score shows the accuracy of the test, which is appropriate when working with imbalanced data [40]. It can be intuitively understood that higher TPR and NRR represent better classification ability. Table 1 shows the classification accuracy of symptom description domain partition. Intuitively, the normal ECG can be very accurately separated from other heartbeats with pathological features.

To show the classification results of each specific disease and the validity of symptom description domain partition, we use principal component analysis (PCA) to obtain the confidence elliptic region [41], as shown in Figure 12. Notice that almost every confidence elliptic region is contained in the corresponding symptom description domain, hence the selection of our symptom description domain partition has strong robustness.

Although the results from normal ECG signals are basically identical, ECG signals with different pathological features tend to have different degrees of change and ECG signals may vary even if they share the same pathological feature. Thus, firstly, we infer the location of cardiac abnormalities such as the atrium, ventricle and atrioventricular bundle. Secondly, we give the reference diagnostic results. For those ECG signals in which we can infer the location of cardiac abnormality but cannot be sure of the exact disease, doctors need to make further diagnoses. Furthermore, we can estimate the severity of pathology according to the longitudinal dispersion.

### 4.2. Comparation with PH Method and CNN Method

Table 2 shows the classification result of pathological classes by PH method. The main idea is to utilize persistent homology to construct a topological structure to classify point clouds after STFFT. It is noted that the PH method can characterize the normal and pathological ECGs, but the performance of classifying different pathological ECGs is not satisfactory. This may be because the topological method largely reflects the overall feature of the point cloud and ignores the local difference. However, the WSCEC algorithm emphasizes local structure and makes up for this defect.

Table 3 shows the classification result of pathological classes by CNN method. We sample 1500 ECG signals as the training dataset and set the other 3500 ECG signals as the test database. The overall accuracy of the CNN method is similar to the WSCEC method. It is noted that $F_{1}$ score of atrial abnormality in CNN is lower than WSCEC, which may be because the training dataset of atrial abnormality is small. This observation reflects the advantage that our WSCEC method does not depend on training samples. As the training samples increase, such as ventricular abnormal and bundle branch block, the accuracy of the CNN method becomes higher, which is a little bit better than WSCEC.

## 5. Conclusions, Limitations and Future Works

In this paper, a novel ECG-assisted classification algorithm based on Wasserstein scalar curvature is proposed. By introducing Wasserstein scalar curvature, the WSCEC algorithm more accurately describes the neighborhood information differences of point clouds obtained after STFFT and shows its ability to classify heart-healthy conditions, especially to recognize atrial fibrillation and ventricular fibrillation. Meanwhile, WSCEC presents its potential in predicting the developing tendency of heart diseases, which might decline the incidence of sudden death. Consequently, a well-packaged heart disease prediagnosis system might be designed.

WSCEC algorithm is an original attempt to incorporate geometric invariants into medical research, however, there remain some limitations. More precise classification of heart diseases in each pathological class needs to be studied. In addition, we will verify the effectiveness of the WSCEC method on many other available ECG databases including more different arrhythmias in the future.

For further research, we hope to combine Wasserstein scalar curvature, topological features and machine learning to comprehensively investigate the signal and obtain more efficient algorithms. Besides, the standard 12-lead ECG auxiliary diagnosis is expected to be studied. In addition, the WSCEC algorithm can be further applied to a variety of signal research, such as signal identification and Electroencephalogram (EEG) analysis. We believe that the geometrical method will show more potential in data science.

## Figures and Tables

**Figure 1 entropy-24-01450-f001:**
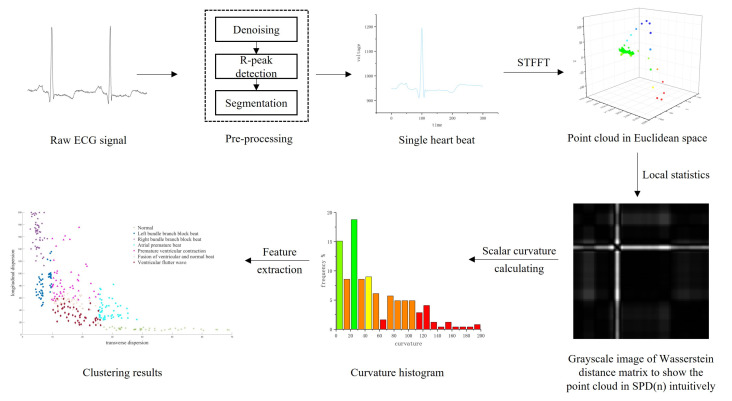
Pipeline of Wasserstein scalar curvature (WSCEC).

**Figure 2 entropy-24-01450-f002:**
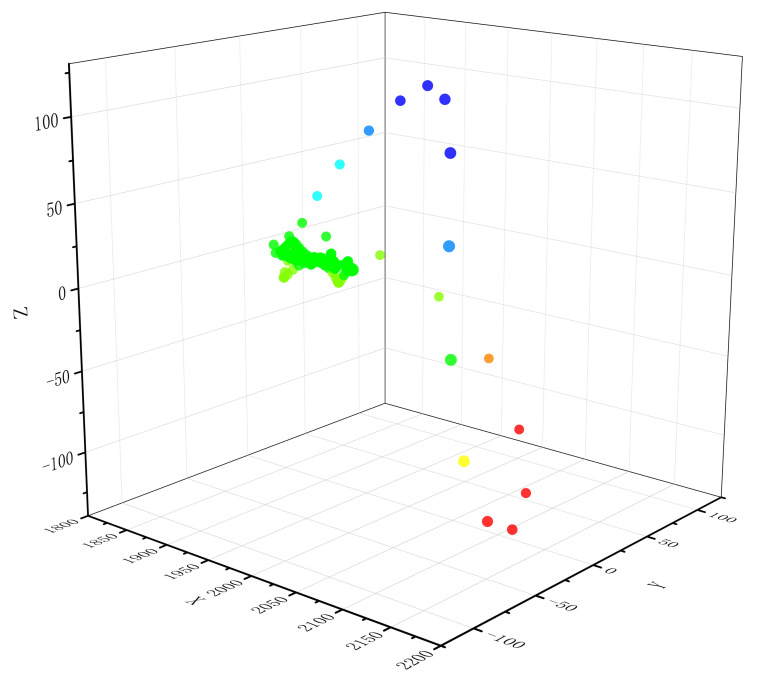
Point cloud of $T_{s}$ in $R^{3}$ after fast Fourier transform (FFT) embedding. We only consider the first three coefficients when using the FFT and sliding windows (we slide the window for fixed length and the Fourier spectrum is then calculated from each window) for visualization.

**Figure 3 entropy-24-01450-f003:**
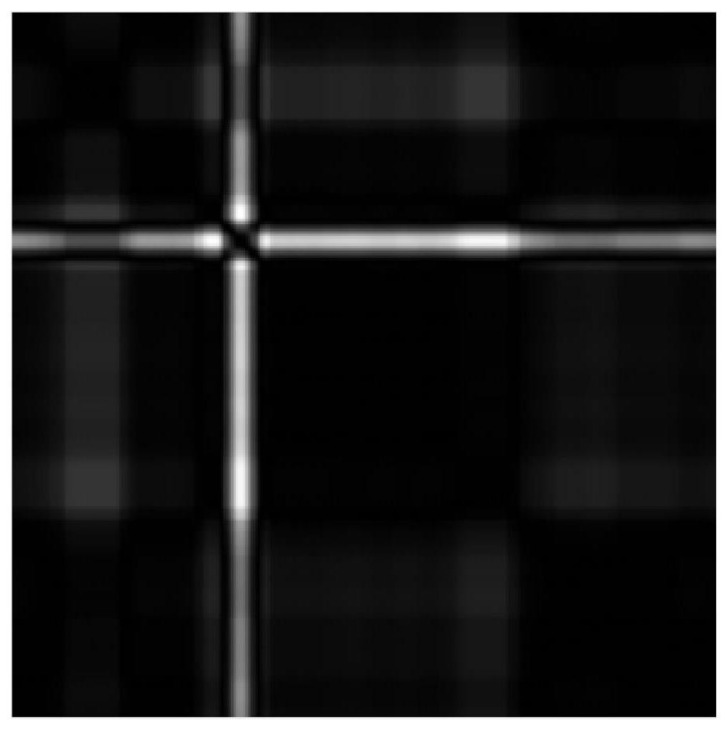
Grayscale image of Wasserstein distance matrix for point cloud of $T_{s}$ on SPD(3). Wasserstein distance helps to understand the difference between two local structures in a point cloud after STFFT.

**Figure 4 entropy-24-01450-f004:**
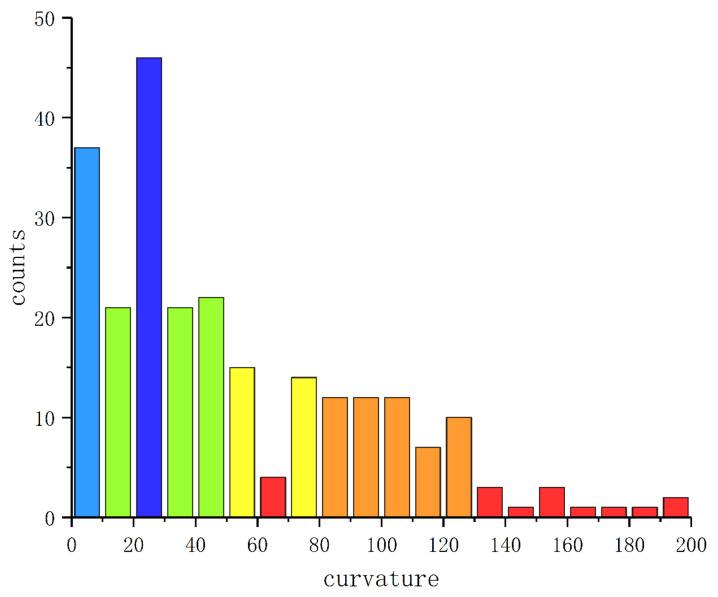
Wasserstein scalar curvature histogram (WSCH) for distribution point cloud of $T_{s}$.

**Figure 5 entropy-24-01450-f005:**
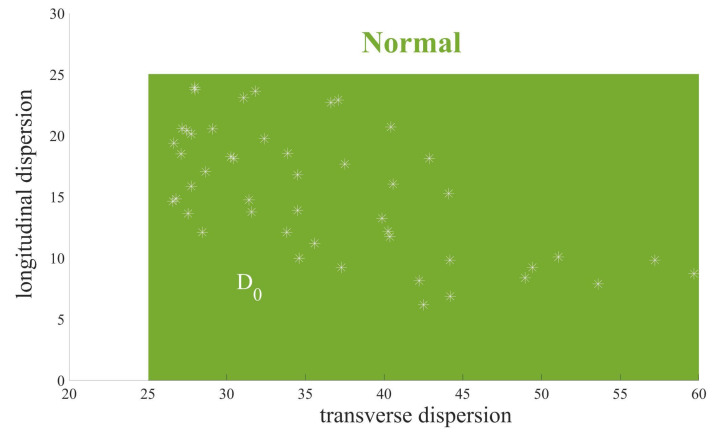
Wasserstein scalar curvature dispersion (WSCD) of normal ECG signals. WSCD of healthy heartbeats are more even because of the strong regularity.

**Figure 6 entropy-24-01450-f006:**
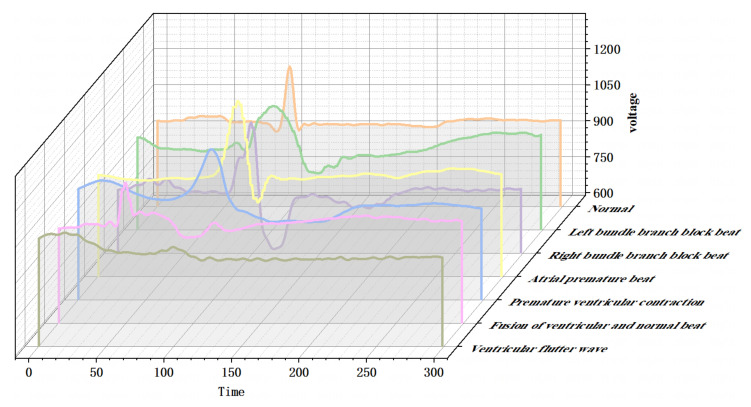
Schematic diagram of seven kinds of Electrocardiograms (ECG) signals.

**Figure 7 entropy-24-01450-f007:**
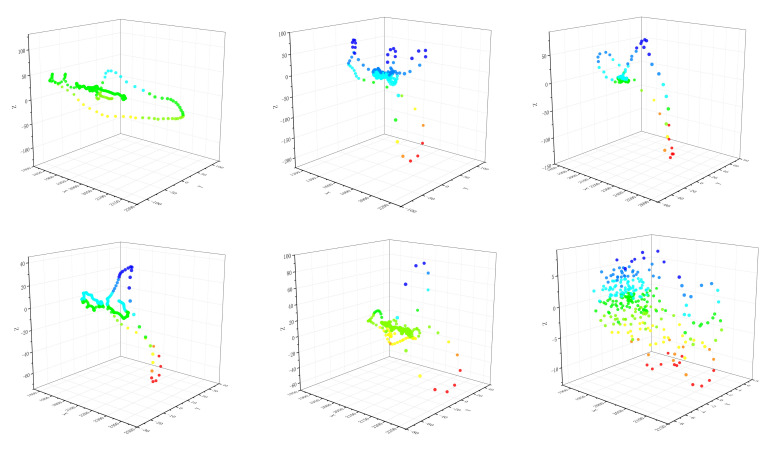
Point clouds of ECG signals with six pathological features in $R^{3}$ after STFFT. The first row from left to right represents left bundle branch block heartbeats (L.B.B.B.), right bundle branch block heartbeats (R.B.B.B.), atrial premature heartbeats (A.P.), respectively, and the second row from left to right represents premature ventricular contraction heartbeats (P.V.C.), ventricular and normal heartbeats (F.V.N.), ventricular flutter heartbeats (V.F.), respectively.

**Figure 8 entropy-24-01450-f008:**
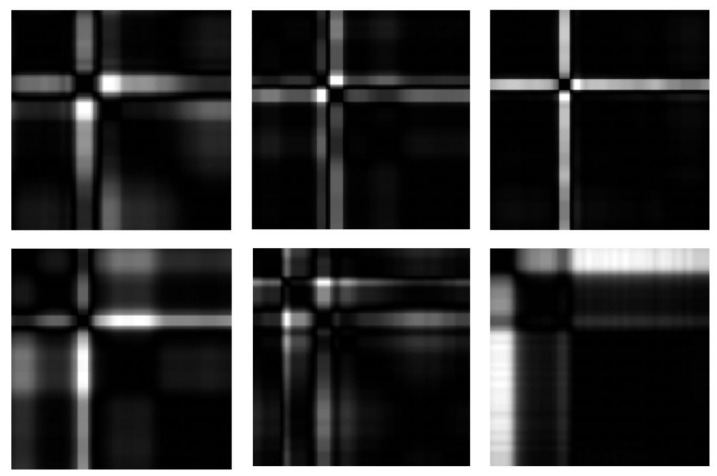
Grayscale image of Wasserstein distance matrices for distribution point clouds. The first row from left to right represents L.B.B.B., R.B.B.B., and A.P., respectively, and the second row from left to right represents P.V.C., F.V.N., and V.F., respectively.

**Figure 9 entropy-24-01450-f009:**
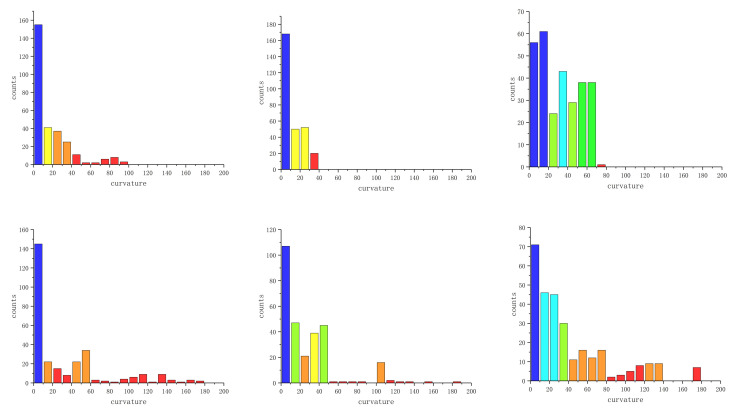
WSCHs of distribution point clouds of six pathological ECG signals. The color in WSC histograms means the difference in the size of each column (the higher the column, the purplier the color, and the lower the column, the redder the color). The first row from left to right represents L.B.B.B., R.B.B.B., and A.P., respectively, and the second row from left to right represents P.V.C., F.V.N., and V.F., respectively.

**Figure 10 entropy-24-01450-f010:**
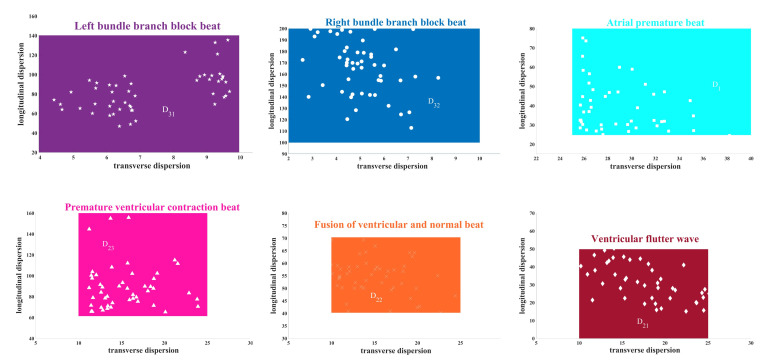
WSCD of seven kinds of ECG signals. The first row from left to right represents L.B.B.B., R.B.B.B., and A.P., respectively, and the second row from left to right represents P.V.C., F.V.N., and V.F., respectively.

**Figure 11 entropy-24-01450-f011:**
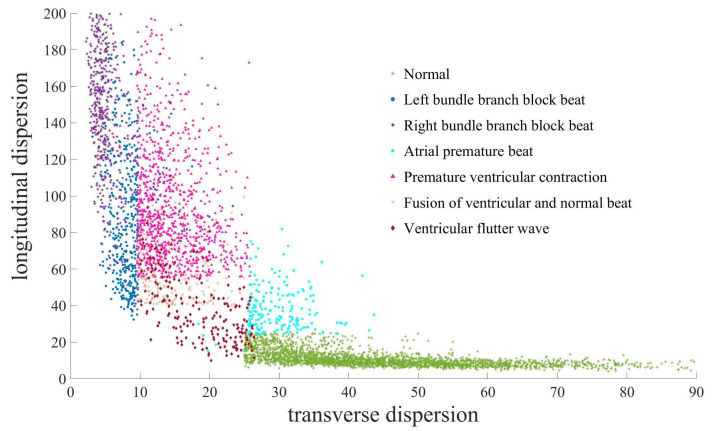
Wasserstein scalar curvature dispersion of seven kinds of ECG signals.

**Figure 12 entropy-24-01450-f012:**
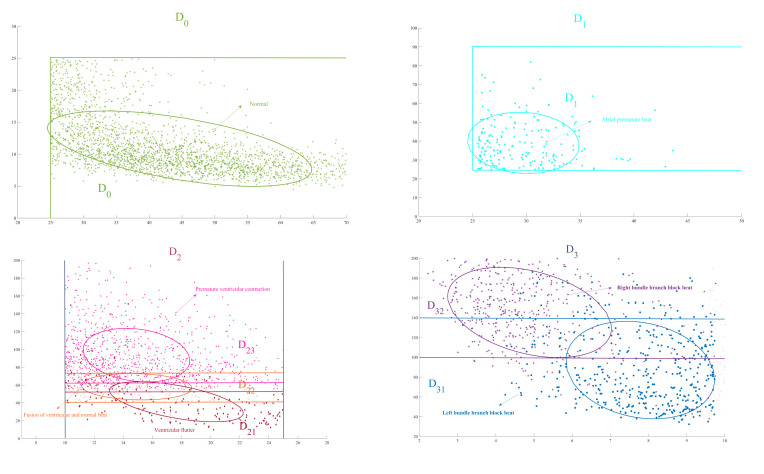
Principal component analysis result. Every confidence elliptic region is almost contained in the corresponding symptom description domain.

**Table 1 entropy-24-01450-t001:** Classification result of symptom description domain partition with WSCEC.

ECG Type	Normal	Atrial Abnormal	Ventricular Abnormal			Bundle Branch Block		Unclassified
	N	A. P.	P. V. C.	F. V. N.	V. F.	L. B. B. B.	R. B. B. B.	Null
Original size	2500	200	1400			900		0
Classified size	2513	212	1305			970		0
TPR	99.96%	95.50%	91.36%			97.78%		0
NRR	99.44%	99.56%	99.35%			97.80%		0
PPV	99.44%	90.09%	98.01%			90.72%		0
$F_{1}$ score	99.70%	92.72%	94.57%			94.12%		0

**Table 2 entropy-24-01450-t002:** Classification result of symptom description domain partition with PH method.

ECG Type	Normal	Atrial Abnormal	Ventricular Abnormal			Bundle Branch Block		Unclassified
	N	A. P.	P. V. C.	F. V. N.	V. F.	L. B. B. B.	R. B. B. B.	Null
Original size	2500	200	1400			900		0
Classified size	2980	530	561			929		0
TPR	97.84%	11.50%	35.64%			45.67%		0
NRR	78.64%	89.44%	98.28%			87.37%		0
PPV	82.08%	4.34%	88.95%			44.24%		0
$F_{1}$ score	89.27%	6.30%	50.89%			44.94%		0

**Table 3 entropy-24-01450-t003:** Classification result of symptom description domain partition with CNN method.

ECG Type	Normal	Atrial Abnormal	Ventricular Abnormal			Bundle Branch Block		Unclassified
	N	A. P.	P. V. C.	F. V. N.	V. F.	L. B. B. B.	R. B. B. B.	Null
Original size	1727	140	1019			614		0
Classified size	1728	155	992			625		0
TPR	99.76%	87.14%	93.62%			96.57%		0
NRR	99.71%	99.01%	98.47%			98.30%		0
PPV	99.71%	78.70%	96.17%			94.88%		0
$F_{1}$ score	99.73%	82.71%	94.88%			95.72%		0

## Data Availability

Not applicable.

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
