# Peer review of "ECG Classification Based on Wasserstein Scalar Curvature"

_entropy, 2022, doi:10.3390/e24101450_

Round 1

Reviewer 1 Report (Previous Reviewer 1)

The authors have included all the required points in the new version of the manuscript.

Author Response

Dear Reviewer,

We deeply appreciate the time and effort you’ve spent in reviewing our manuscript named as ” ECG Classification based on Wasserstein Scalar Curvature” with Manuscript ID: entropy- 1944202. Thanks for the insightful comments and suggestions again.

With best regards,

Huafei Sun

Reviewer 2 Report (Previous Reviewer 3)

My comments have been addressed, so I recommend acceptance. 

Author Response

Dear Reviewer,

We deeply appreciate the time and effort you’ve spent in reviewing our manuscript named as ” ECG Classification based on Wasserstein Scalar Curvature” with Manuscript ID: entropy- 1944202. Thanks for the insightful comments and suggestions again.

With best regards,

Huafei Sun

Reviewer 3 Report (Previous Reviewer 2)

Thank you for the response to my previous comments. However, only some of them was addressed in the manuscript itself. All the questions should be answered in the manuscript, not only in the message to the reviewer. Main issues were not addressed at all. For example, you did not proof the benefits of WSC as compared to using raw ECGs or FFT coefficients. I do not agree, that there is limited number of studies, where morphological/rhythmic ECG features were used to classify the same number of the arrhythmias. In last decade, there was a lot of similar studies, using or not using machine learning techniques (it is not the question in this case). Yes, machine learning methods have often the character of a black box. Your features are, however, also a black box for the readers, as in the manuscript you did not explain the relationship between the FFT clouds, initial raw ECGs, and final WSC metrics. There are some differences in all maps in the manuscript. Bud you could see the similar differences, even if you plot 3D cloud from three raw ECG channels (even if you will draw raw ECG clouds). So, they can ask you again: what is the beneficial of using the curvature metrics instead of „really“ simply interpretable ECG features? To proof your suggestions about your method advantages, you should provide the comparison with another methods (even if they will be based on using feature-based machine learning techniques, more or less interpretable) or carry out the experiments, where you will use different ECG representation (for example raw ECG cloud vs. FFT cloud, etc.) for arrhythmia classification in order to verify, whether the proposed method is the best against other similar (and more simple) methods. You could experiment with data, as many other available ECG databases (including more than 20 different arrhythmias) have been recently introduced. Practical application of the proposed method is as such disputable, as use of deep learning black boxes, as doctors will not be familiar with the features (WSC) you use to classify arrhythmias. Thus, the research you made is not comprehensive enough, as compared to other previous and recent reports. Unfortunately, I still cannot recommend the manuscript to publication.

Author Response

Dear Reviewer,

We deeply appreciate the time and effort you’ve spent in reviewing our manuscript named as ” ECG Classification based on Wasserstein Scalar Curvature” with Manuscript ID: entropy- 1944202. Thanks for the insightful comments and suggestions again. According to your suggestions, we have given the revises in the attachment.

With best regards,

Huafei Sun

Round 2

Reviewer 3 Report (Previous Reviewer 2)

The manuscript was enriched by comparison with some another techniques, so, the results of original method are now more valid than in older versions. Please, revise text, which was added in last version (grammar and typing errors). Thank you for your effort. Wish you success in future research!

Author Response

Dear Reviewer, 

Thanks for your effort and suggestions. We have revised the typos in the latest version.

With best regards,

Huafei Sun

This manuscript is a resubmission of an earlier submission. The following is a list of the peer review reports and author responses from that submission.

Round 1

Reviewer 1 Report

The paper ''ECG Classification based on Wasserstein Scalar Curvature” presents a methodology based on the optimal transport related metric named Wasserstein distance to analyze and to classify heart-healthy conditions. The topic is of interest for data scientists and especially for doctors. I think the paper is in principle interesting, but it still needs to be improved. For example, the Introduction of the manuscript is poor, lacking a brief review of the state-of-the-art regarding the Wasserstein distance in various contexts, since it is very useful in physics through optimal transport problems. Also, the presentation is confusing, with missing or poorly motivated definitions (see comments below). In fact, crucial definitions like "PQRST complex" are missing. The use of English may be improved, since there are spelling and grammatical mistakes that should be corrected (some examples will be given below). 

Furthermore, the manuscript needs major improvement in some aspects: 

(1) Line 26: What does “normal ECG” mean? Please clarify. 

(2) Lines 26-29: To reach a wide audience, please explain what the "PQRST complex" is in the Introduction section.

(3) Lines 26-32: From Reference 3 jumped to Reference 14, is this correct?

(4) "Fast Fourier transform" (for instance in line 48) or "fast Fourier transform" (for instance in line 64, after the subsection title), please choose one of the two and keep throughout the text. I suggest the second one.

(5) Please double-check equation 2. Who is capital N? I assume the correct is lowercase n. Also, emphasize that a_k and b_k are definite positives. Otherwise, equation 2 is not always true.

(6) Regarding the abbreviations, they must be inserted in parentheses right after the written-out means, when defined for the first time. This one should be made at its first appearance in the abstract, in the first figure (or table), and in the main text (please see the "Instructions for Authors" in https://www.mdpi.com/journal/entropy/instructions for more details). I suggest that the authors review the abbreviations entered in the manuscript. Below, I list some suggestions.

> Line 19:  Please change “ECG” to “Electrocardiograms (ECG)”.  

> Line 26: What does “PQRST” mean? Please enclose the meaning of PQRST and put this abbreviation in parentheses on this line. 

>Line 75: Please change “kNN algorithm” to “ k-nearest neighbors (KNN) algorithm”.

> Figure 1.: Please change the caption “Pipeline of WSCEC” to “Pipeline of Wasserstein scalar curvature (WSCEC)”. The same rule for FFT in Figure 2, WSCH in Figure 4, among others.

(7) Line 79: To show the importance of this metric, please reference some more recent work, such as:

> (Machine Learning)You, C.; Yang, J.; Chapiro, J.; Duncan, J.S.; (2020). Unsupervised Wasserstein Distance Guided Domain Adaptation for 3D Multi-domain Liver Segmentation. In: Interpretable and Annotation-Efficient Learning for Medical Image Computing. IMIMIC MIL3ID LABELS 2020. Lecture Notes in Computer Science, 12446. Springer, Cham.

> (Physics) Koehl, P.; Delarue, M.; Orland, H.; (2019) Statistical Physics Approach to the Optimal Transport Problem, Phys. Rev. Lett., 123(4) 040603.

> (Geophysics) da Silva, S.L.; Karsou, A.; de Souza, A.; Capuzzo, F.; Costa, F.; Moreira, R.; Cetale, R.; (2022) A graph-space optimal transport objective function based on q-statistics to mitigate cycle-skipping issues in FWI, Geophys. J. Int,, 231(2) 1363-1385.

> (Information theory) Zhang, Y.; Zhao, Z.; Deng, Y.; Zhang, X.; (2022) FHRGAN: Generative adversarial networks for synthetic fetal heart rate signal generation in low-resource settings, Inf. Sci., 594, 136-150.

(8) Lines 79-80: Wasserstein distance is defined for the expected value of any cost function. What the authors show in equation 7 is actually the Wasserstein distance based on the Lp norm. Please mention that equation 7 is the Wasserstein distance based on the Lp norm and cite book of the Dr. Cédric Villani to present this equation.

> Villani, C., 2008. Optimal Transport: Old and New, Grundlehren der mathematischen Wissenschaften, Springer Berlin Heidelberg.

(9) Eq. 8: Make it clear what value of p was considered in equation 8?

(10) Please double-check the references. Below, I give you some examples.

> Reference 1.: Websites may change or disappear over time. Thus, it is important to inform the date of the last access performed by the authors, as follows:

Now: 1. World Health Qrganization. The top 10 causes of death.  https://www.who.int/news-room/fact-sheets/detail/287 the-top-10-causes-of-death

Suggested: 1. World Health Organization. The top 10 causes of death. Available online: https://www.who.int/news-room/fact-sheets/detail/287 the-top-10-causes-of-death (accessed on Day Month Year).

Please also correct the word "Q"rganization.

> Reference 2.: I cannot access Reference 2. Can the authors provide a better description of this one? A website, DOI link, etc.

In my opinion, the article should address the above points in order for the work to be interesting for several readers. The above points must be addressed before the manuscript can be considered for publication.

Author Response

Dear Reviewer,

We deeply appreciate the time and effort you’ve spent in reviewing our manuscript named as ” ECG Classification based on Wasserstein Scalar Curvature” with Manuscript ID: entropy-1882927. Thanks for the insightful comments and suggestions again. According to your suggestions, we have given the revises as follows.

With best regards,

Huafei Sun

Reviewer 2 Report

The proposed approach is highly complex (a lot of steps when calculating the features) and non-intuitive. The description is based on the math background with no sufficient explanation of the relationships between used classification features and analysed ECGs. The benefits against other feature-based or deep-learning-based methods are not stated at all. The comparison to other report is completely missing. Method is tested on ECG with morphological manifestations of arrhythmias only. Selected pathologies are relatively accurately separated using simple features, such as QRS duration and amplitude, P wave direction, etc. The practical applicability of such poorly intuitive method is, therefore, discutable.  It is hard to predict, whether it would be beneficial when recognizing poorly differentiated arrhythmias, such as atrial flutter, atrial fibrillation, etc. Although the authors did a lot of work, there is no strong evidence of the usability of the proposed method. Therefore, I cannot recommend the manuscript to publish in Entropy.

Interpretation of the approach steps and outcomes is insufficient. Describe/interpret „the point cloud“ calculated using FFT (see Fig.2). What does the samples and the cloud „structure“ mean in context of initial raw ECG segment? What does three dimensions in the Figure mean in this context (what type of relationship do the points represent)? Explain the Wasserstein distance „map“ in Fig. 3. What does it represent in terms of initial ECG segment and FFT embedding points?

It would be interesting to compare the results obtained by using WSCH features and point clouds in order to verify, whether complex WSCH feature calculation is beneficial as compared to simple method based on FFT representation (e.g. Fig.7). The results should be compared with other reports on the same database. The pros and cons of the proposed method should be clearly stated and discussed in context of real practical applications.

The classification performance should be assessed by standardly used sensitivity, positive predictive value, F1-score, etc. F1-score is appropriate when working with imbalanced data (see Table 1).

Graphical illustrations should be improved. The axes in the figures should be appropriately named, including the units. What does the color in WSC histograms mean (e.g. Fig. 4)? Y axis in Fig.9 should be of the same range in order to properly compare the histograms obtained for different ECGs. Axes in Fig.10 should be of the same range in order to compare the results obtained for different ECGs.

Setting of some „hyperparameters“ of the method should be explained in detail. What does „length 300“ mean (see Preprocessing of ECG signal)? Is this in seconds, samples, …? How did you choose the segment duration? For different segment types (e.g. normal and LBBB or LBBB), appropriate length can differ significantly. Why did you set „k“ in kNN on 20 points? What does „original size“ and „classified size“ mean in Table 1? Calculate classification performance metrics for each pathology separately instead of 4 presented groups.  Add the main performance metrics in Abstract. Rephrase „the symptom description domain partition“.

Finally, your „FFT embedding“ method is usually called short-time Fourier transform in signal processing area (the long signal is devided into the segments and the Fourier spectrum is then calculated from each segments). If I am right, and you use STFT method, then please use this, standardly used name in the method description. Spectral components and, consequently, features, depend on the spectral resolution. What were the parameters of FFT and spectra in your case?

Author Response

(The authors gave the same response as above.)

Reviewer 3 Report

Paper title: ECG Classification based on Wasserstein Scalar

Curvature

There are some points need to be further clarified:

1-       The presentation and structure are not good and thus prevent me from understanding the novelty and measuring its quality.

2-       The motivation on the study should be further emphasized, particularly; the main advantages of the results in the paper comparing with others should be clearly demonstrated. 

3-       The example section needs to be further expanded and including some remarks to show the effectiveness and efficiency of the proposed method, compared with others. 

4-       The experiments and discussion, the number datasets may not show the efficiency of the method as well as the one experiment may not show the efficiency of the method.

5-       The literature review should be extended. You may cite some latest excellent papers of swarm intelligence including as a reference.

Recommendation: According to all these issues,

Decision: major revisions

Author Response

(The authors gave the same response as above.)
